# GLOFs in the WOS: bibliometrics, geographies and global trends of research of glacial lake outburst floods (Web of Science, 1979-2016)

Adam Emmer[1]

[1]Czech Academy of Sciences, Global Change Research Institute (CzechGlobe), Brno, 603 00, Czech Republic

*Correspondence to*: Adam Emmer (aemmer@seznam.cz)

**Abstract.** Research of glacial lake outburst floods (GLOFs) - specific low frequency, high magnitude floods originating in glacial lakes, including jokulhlaups - is well justified in the context of glacier ice loss and glacial lake evolution in glacierised areas all over the world. Increasing GLOF research activities, which are documented by the increasing number of published research items, have been observed in the past few decades; however, a comprehensive insight into the GLOF research

community, its global bibliometrics, geographies and trends in research is missing. To fill this gap, a set of 892 GLOF research items published in the Web of Science database covering the period 1979-2016 was analysed. General bibliometric characteristics, citations and references were analysed, revealing a certain change in the publishing paradigm over time. Furthermore, the global geographies of research on GLOFs were studied, focusing on: (i) where GLOFs are studied; (ii) who studies GLOFs; (iii) the export of research on GLOFs; and (iv) international collaboration. The observed trends and links to

the challenges ahead are discussed and placed in a broader context.

## 1 Introduction

*Glacial lake outburst flood (GLOF)* is a term used to describe a sudden release of (part of) water retained in a glacial lake, irrespective of the cause (trigger), mechanism (dam failure or dam overtopping) and glacial lake sub-type involved (e.g., Evans and Clague, 1994; Richardson and Reynolds, 2000). The Islandic term *jokulhlaup* is also frequently used to refer to GLOFs

originating in ice-dammed lakes and often (but not necessarily) induced by volcanic activity (Björnsson and Pálsson, 2008). In Spanish speaking countries, the term *aluvión* is often used to described flow-type processes of various origins, including GLOFs (Lliboutry et al., 1977). The occurrence of GLOFs is commonly tied to periods of glacier ice loss (Clague and Evans, 2000) and GLOFs are considered among the most significant geomorphological as well as the most hazardous consequences of retreating glaciers (Clague et al., 2012). Facing ongoing and even accelerating climate change and the associated glacier

retreat (Huss et al., 2017), research of past as well as potential future GLOFs represents a highly actual topic (O'Connor and Clague, 2015). A complex understanding of GLOFs and their related processes has significant implications for various fields such as risk management and disaster risk reduction (e.g., Hewitt, 2013), hydropower plant design (e.g., Schwanghart et al., 2016), or geomorphology and sediment yield (e.g., Korup and Tweed, 2007).

According to the Web of Science (WOS) Core Collection database (see 2.1), the term *glacial outburst flood* was firstly used by Jackson (1979), the term *glacial lake outburst flood* was firstly used by Clarke and Mathews (1981) and the acronym *GLOF* was firstly used by Grabs and Hanisch (1993). The history of research on GLOFs is, however, much older, going back to the 19th century in some regions, e.g., the European Alps (e.g., Richter, 1892), Iceland (e.g. Hákonarson, 1860) and the first half of the 20th century in others, e.g., Caucasus (Gerassimow, 1909), the Peruvian Andes (e.g., Broggi, 1942) or the North American Cordillera (e.g., Fryxell and Horberg, 1943).

Numerous studies exist that focus on a scientometric / bibliometric analysis of various research fields and their geographies (e.g., Small and Garfield, 1985), including those in geomorphology (e.g., Dorn, 2002), river research (e.g., Vugteveen et al., 2014), and natural hazard science (e.g., Chiu and Ho, 2007; Liu et al., 2012; Wu et al., 2015). While the amount of research of GLOFs has gradually increased over the past few decades (see Emmer et al., 2016), detailed insight into the GLOF research community, its global bibliometrics and geographies of research is missing. Therefore, the main objective of this study is to map GLOF research and the research community by analysing GLOF research items published in the WOS Core Collection database from the point of view of: (i) general bibliometric and scientometric characteristics (temporal analysis, journals, document types, citations and references); (ii) geographies (where GLOFs are studied and by whom in relation to where the GLOFs occur). This study is expected to provide a comprehensive global overview primarily targeting the GLOF research community, but also a broader audience such as GLOF risk management practitioners and policymakers (see e.g., Carey et al., 2012).

## 2 Data and methods

### 2.1 Web of Science and the set of analysed items

Focussing on research results published in highly credible scientific journals and proceedings, the search was performed on items published in journals/proceedings indexed in the Clarivate Analytics Web of Science (WOS) Core Collection database (www.webofknowledge.com). This database covers items published as far back as 1900 in 13,000+ journals, contains 100+ million published items and 1.3 billion cited reference connections (Clarivate Analytics, 2017). Numerous books (e.g., Carey, 2010; Haeberli and Whiteman, 2015), book chapters (e.g. Iturrizaga, 2011) and papers not indexed in the WOS Core Collection databases as well as so called grey literature (e.g., Horstmann, 2004) are not taken into consideration in this study. Also, various scientific reports of national and regional authorities such as the ICIMOD in Hindu Kush-Himalaya region (e.g., Ives et al., 2010), USGS in the North American Cordillera (e.g., O'Connor et al., 2001), or the National Water Authority in the Peruvian Andes (e.g., UGRH 2015) are not taken into consideration due to the often limited availability (hardcopies stored in local archives; no electronic versions available) and unbalanced overall amount of these sources among the regions. Moreover, these data are often not easily describable by standardized WOS characteristics and, thus, difficult to analyse as a comprehensive set of items on a global level. Such studies and the employment of documentary data, however, may provide a valuable source of information for analysing the occurrence, causes and mechanisms of GLOFs on a regional level (e.g., Emmer, 2017) and,

in turn, to facilitate the compilation of a GLOF database on a global level (see also 4.3). Nevertheless, an analysis of these studies is out of the scope, extent and resources of this study. The WOS database was, therefore, used as the only source of data and the presented results are, thus, only valid among the items indexed in the WOS.

The proper definition of the search formula is a key presumption for successfully searching for a comprehensive set of items. After several iterations, sighting shots and cross-checks with the control set of 30 papers focusing on diverse aspects of research of GLOFs, a basic searching formula was defined as follows:

TOPIC: (glaci* AND outburst* AND flood* OR jokulhlaup*)

This formula was defined to cover the terminological variability observed in this field over time and lake types, and simultaneously omit undesirable results such as items focusing on outburst floods from landslide-dammed lakes. The WOS database was analysed between September and October 2017, when all of the major journals regularly publishing GLOF research items have already released their 2016 issues to WOS database. Each item contained in the WOS Core Collection database is described by a number of qualitative and quantitative characteristics, some of which are further analysed in this study (Tab. 1; see Section 3), reflecting the objectives. A total of 892 items were found and used for the building and analysis of the database (see 2.2).

## 2.2 Data processing, database building and analysis

The basic analysis was performed in the WOS environment, using the WOS Results Analysis tool. This tool allows the analysis of the entire set of 892 items, including an overview of: (i) authors; (ii) institutions; (iii) source titles; (iv) WOS categories; (v) research areas; (vi) document types (see 3.1). Journals publishing GLOF research items (see 3.1.1) were further analysed using the WOS Journal Citation Reports tool and citations and references (see 3.1.2) were analysed using the WOS Citation Report tool. The set of 892 analysed GLOF research items (see 2.1) was exported from the WOS platform, and subsequently processed and analysed in a Microsoft Excel environment. Some of the data needed to be repaired (e.g., the institutions of all of the authors involved in each research item are not available for 21 (2.4%) research items written by more than one author; in which case, the country of the affiliation of the reprint author was used).

In addition to the characteristics assigned from the WOS database, each item was manually described by certain other characteristics, based on the search in the titles, abstracts (available for 843 out of the 892 items; 94.5%), and keywords (available for 517 out of the 892 items; 58.0%), using the content analysis method (e.g., Hsieh and Shannon, 2005). These include geographical characteristics, characteristics of authors (teams) contributing to individual research items and mono-/inter-nationality of the research teams. These characteristics are analysed in Section 3.2, focusing on: (i) geographical focus of GLOF research (see 3.2.1); (ii) geographies of the researchers involved (see 3.2.2); (iii) the "export" of research of GLOFs (see 3.2.3); and (iv) international collaboration (see 3.2.4). The content of GLOF research items is further analysed regarding selected aspects of their focus (see 3.3), using the WOS Advanced Search tool. This analysis is based on searching among the

titles, abstracts and keywords and, thus, only provides a basic, nonetheless meaningful view on the given aspect. The observed trends are discussed and placed into a broader context in Section 4.

## 3 Results

### 3.1 General characteristics of published items

#### 3.1.1 Bibliometric characteristics

A total of 892 GLOF research items were published in the WOS Core Collection database as of the end of 2016. At least one GLOF research item was published each year since 1979 with the exception of 1983 and 1987 (see Fig. 1). At least 10 GLOF research items were published every single year since 1993, at least 20 items since 2002 and at least 50 items since 2012. A record number of papers was published in 2015 (n = 81; 9.1%). Nearly 500 items (n = 498; 55.8%) have been published since
2008 (during the past 8 years out of the 38-year long period of the analysis).

GLOF research has been published under 256 diverse *source titles* (journals and proceedings) indexed in the WOS, of which 26 journals / proceedings have published 10+ GLOF research items each (515 items in total; 57.7%) and 10 journals / proceedings have published 20+ items each (297 items in total; 33.3%). The Journal of Glaciology (n = 48; 5.4%), Quaternary Science Reviews (n = 46; 5.2%) and Geomorphology (n = 39; 4.4%) have published 30+ items, in total 133 (14.9%). Twenty
of the most popular journals / proceedings have published a total of 449 (50.3%) items. Twenty-two out of the 26 journals that published 10+ items each have a Q1 or Q2 ranking in at least one WOS category. Four of these journals have IF > 4, 11 have IF > 3 and 20 have IF > 2. Eight research items (0.9%) have been published in top ranked journals (Nature, Science).

The analysed GLOF research items have been publish in journals indexed in 48 diverse *WOS categories*. The vast majority of published items (n = 558; 62.6%) have been published in journals indexed in the *WOS category* "geoscience
multidisciplinary", followed by "geography physical" (n = 338; 37.9%), "water resources" (n = 149; 16.7%) and "geology" (n = 117; 13.1%). Research of GLOFs is represented in 34 diverse research areas. The most frequently represented *research area* is "geology" (n = 666; 74.7%), followed by "physical geography" (n = 338; 37.9%), "water resources" (n = 149; 16.7%) and "environmental science ecology" (n = 100; 11.2%).

#### 3.1.2 Citations and references

The 892 published GLOF research items have attracted a total of 18,570 citations (as of the end of 2016) and around 13,000 citations without self-citations. The number of citations has gradually increased during the analysed period, from tens of citations per year in the 1980s, hundreds of citations per year in the late 1990s and 2000s, thousands of citations per year in the 2010s, to up to 2,514 citations in 2016, which basically reflects the increasing number of GLOF research items published (see 3.1). Each item has an average of 51.63 records in the list of references, rarely exceeding 100 references.

Each GLOF research item has obtained an average of 20.82 citations (14.57 without self-citations). The most cited paper (Hemming, 2004) has obtained 581 citations (as of the end of 2016) and the 108 most cited papers (12.1% of all) have obtained more than 50% of all citations. Approximately one quarter (n = 228; 25.6%) of all items have two or less citations, of which 107 were published in 2014 or earlier. The H-index of GLOF research items is 62 (October 2017). In terms of citations obtained by individual items per year, 7 items obtained 15+ citations / year, 21 items obtained 10+ citations / year, and 77 items obtained 5+ citations / year during the analysed period. Each item obtained an average of 2.11 citations (1.48 excluding self-citations) per publication–year (8,798 publication-years in total). The ratio of citations in a given year divided by the cumulative number of items published until a given year has gradually increased from 0.28 in the early 1990s to 2.90 in 2015.

## 3.2 Geographies of GLOFs, GLOF research and researchers

### 3.2.1 Where are GLOFs studied?

The occurrence of GLOFs is closely tied to the retreating glaciers and is, therefore, geographically clustered. Based on the previous GLOF inventories and overviews (Carrivick and Tweed, 2016; Emmer et al., 2016), eleven non-overlapping hotspots of GLOFs occurrence are distinguished around the globe (see Tab. 2; Fig. 2), of which three are located in Asia (HKH, CAS, KRK), in Europe (ALP, ICL, SCA) and North America (ALA, GRL, NAC) and two are located in South America (CAN, PAN). ICL, NAC and HKH are the most prominent GLOF research hotspots with 180, 144 and 142 research items, respectively. The documented numbers of GLOFs are, however, highly disproportionate (see 4.2). GLOFs have been studied on all of the continents including Africa (e.g. Girard et al., 2012), Australia (e.g., Goodsell et al., 2005) and Antarctica (e.g., Margerison et al., 2005), and extreme outburst floods are also thought to shape the landscape of Mars (e.g., Lapotre et al., 2016).

Three types of studies are distinguished among the GLOF research items in terms of geographical focus: (i) regionally-focused items; (ii) multi-regionally-focused items; and (iii) items with no regional focus (e.g., theoretical or model studies). It was determined that almost three quarters of all GLOF research items (n = 651; 73.0%) have their geographical focus in 11 of the pre-defined hotspots of GLOFs occurrence, of which 32 items (3.6%) are characterised as multi-regionally-focused (e.g., Worni et al., 2012; Haeberli et al., 2016). However, the actual number of research items with a certain geographical focus is slightly higher (> 80%), considering items that are geographically focused on regions outside the eleven hotspots of GLOF occurrence, such as Altai / Siberia (21 items, e.g., Margold et al., 2011), Svalbard (5 items; e.g., Schoner and Schoner, 1997), Caucasus (4 items; e.g., Petrakov et al., 2012), Antarctica (21 items, e.g. Margerison et al., 2005) or items focused on Mars (23 items; e.g., Rodriguez et al., 2015). In terms of individual countries, the most prominent country of GLOF research is Iceland (180 research items).

While the total amount of published GLOF research items is gradually increasing globally over time (see 3.1.1, Fig. 1), significant differences exist between individual GLOF research hotspots (see Fig. 3). The amount of research preformed in GLOF research hotspots located in developing regions (e.g., CAS, KRK, CAN, PAN) has generally increased, while the

amount of research performed in developed regions has been stagnant (e.g., SCA, NAC) or has even decreasing (ICL) over the past few decades. The exceptional position of the HKH region in GLOF research can also be seen in Fig. 3. Furthermore, it is clear that GLOFs were studied in hotspots located in Europe and North America before 1991, and expanded to Asia in the 1990s and South America in the 2000s (not considering local publications not indexed in the WOS, see 2.1).

5       From the point of view of the geographical focus on individual lakes, 37 items are focused on repeated Late Pleistocene outbursts from the proglacial Lake Missoula, likely being the most researched glacial lake outburst flood (e.g., Benito and O'Connor, 2003), followed by the 8.2 ka outburst flood from Lake Agassiz, which is mentioned in 28 research items (e.g., Clarke et al., 2004). Various aspects of jokulhlaups on the Katla volcano, Iceland, are elaborated in 27 research items (e.g., Duller et al., 2014). The most famous glacial lakes in GLOF hazard / risk studies nowadays - Imja Tsho, Tsho 10  Rolpa (both Nepal Himalayas) and Palcacocha (Peruvian Andes) have received the attention of 17, 9, and 7 research items, respectively (e.g., Rounce et al., 2016; Klimeš et al., 2016).

### 3.2.2 Who studies GLOFs?

In total, 1,885 authors from more than 750 institutions in 45 countries have contributed to the 892 analysed GLOF research items. Considering the number of items published by each author, it emerges that a research item on GLOF is written on 15  average by 3.49 authors, indicating generally small teams executing GLOF research. Almost one sixth of the 892 research items (n = 146; 16.4%) have been written by individuals, while 746 items (83.6%) by research teams (two or more co-authors). The share of research items written by individuals, however, is dramatically decreasing over time. While 32.8% of all research items published before 2000 were written by individuals, 20.9% were written in 2000-2004, 17.2% in 2005-2009, 6.0% in 2010-2014 and 7.1% in 2015-2016, revealing a certain change in the publication paradigm (see also 4.1). Slightly more than 20  two thirds of GLOF research items (n = 632; 70.9%) have been published by mononational research teams/individuals, while 260 research items (29.1%) have been published by groups of authors from two or more countries (international research teams; see 3.2.4). Researchers from up to six countries have been involved in one individual GLOF research item (e.g., Benn et al., 2012). The share of GLOF research items written by international research teams is gradually increasing over time from 26.4% in 2000-2004, 30.2% in 2005-2009, 35.6% in 2010-2014 to 36.8% in 2015-2016. However, significant differences exist 25  between individual countries (see also Section 3.2.4).

       More than 750 *institutions* have been involved in research of GLOFs, of which 17 institutions have published 20+ GLOF research items (369 items in total; 41.4%). Nine of these 17 institutions are located in Europe (United Kingdom, Iceland, Switzerland and France), seven in North America, and the remaining one in Asia. Overall geographical distribution of GLOF research items by the countries of the authors' institutions suggests a strong dominance of authors from Europe and North 30  America: 497 items (55.7% of all) were (co-)authored by researchers from Europe; 344 items (38.6% of all) were (co-)authored by researchers from North America (USA + Canada); 179 items (20.1% of all) were (co-)authored by researchers from Asia (including Russia); 29 items (3.3% of all) were (co-)authored by researchers from Latin America; 25 items (2.8% of all) were

(co-)authored by researchers from Australia (Australia + NZ); and 7 items (0.8% of all) were (co-)authored by researchers from Africa.

Fourteen out of 1,885 individual researchers (0.7% in all) have published 10+ GLOF research items each (189 items in total; 21.2%) and 90 researchers (4.8% of all) have published 5+ GLOF research items each (436 items in total; 48.9%). Researchers from the USA have contributed to 240 research items (26.9%), followed by 213 items (co)authored by researchers from UK (23.9%) and 115 research items (co)authored by researchers from Canada (12.9%; see Fig. 2). The seven most productive GLOF research countries have published 50+ GLOF research items each (678 items in total; 76.0%) and 16 different countries (England, Scotland and Wales are counted separately) have published 20+ GLOF research items each (802 items in total; 89.9%).

### 3.2.3 The "export" of research of GLOFs

It is obvious that a certain geographical disproportion exists between the countries where GLOFs are studied (11 hotspots of GLOF occurrence; see 3.2.1) and the top 10 GLOF research countries (see 3.2.2; see Fig. 2), which have contributed to a total of 740 (83.0%) GLOF research items. The "export" of research from the top 10 GLOF research countries to 11 hotspots of GLOF occurrence is described in this section (see Tab. 3 and Tab. 4). Considering the country of the first author only (i.e. each item is counted only once in the country of the first author even if it was written by an international research team; Tab. 3), the following patterns are observed: (i) more than three thirds (78.3%) of research items focusing on 11 hotspots of GLOF occurrence were authored by researchers from the top 10 GLOF research countries; (ii) 43.5% of imported research items were authored by researchers from the 10 GLOF research countries; (iii) researchers from the USA and UK have performed research in all 11 hotspots of GLOFs occurrence; (iv) researchers from Iceland, China and Norway have focused almost exclusively on hotspots overlapping with their countries (ICL; CAS, HKH, KRK; and SCA, respectively); (v) the research in ALA and NAC has almost exclusively been performed by the researchers from overlapping countries (74.5% of research items focusing on ALA has been elaborated by researchers from USA and 84.7% of research items focusing on NAC has been elaborated by researchers from USA and Canada).

Considering the countries of all of the institutions contributing to individual GLOF research items (see Tab. 4), it is seen that: (i) researchers from the USA, UK and Switzerland have contributed to research in all 11 hotspots of GLOF occurrence; (ii) the most prominent GLOF research exporter is UK (154 items), followed by USA (84 items) and Switzerland (53 items); (iii) the researchers from Iceland and China focus almost exclusively on hotspots overlapping with their countries (ICL; CAS, HKH, KRK); (iv) the share of imported research from the top 10 countries is very high in GRL, KRK, CAS and HKH, while it is very low in ALP, ALA and NAC; (v) the research in hotspots located in developing countries is dominated by researchers from the USA and Japan (HKH); Germany, Switzerland and UK (CAS); Switzerland, Germany and the USA (KRK), Switzerland and the USA (CAN); and UK (PAN).

Local researchers in hotspots located in developing regions have generally become more active in publishing their research in journals indexed in the WOS over the past few decades (see Fig. 4), which is in line with the increasing amount of

GLOF research items published (see 3.1.1, Fig. 1). The share of items co-authored by local researchers (items written by local and foreign researchers) and authored by local researchers (items written solely by local researchers) varies significantly among the regions. While the share of items authored or co-authored by local researchers is generally increasing (<40% in period 2007-2011 and >50% in 2012-2016), the share of research items published by foreign authors is only slightly increasing in some regions (CAS, PAN). A clear trend is also seen in share of items authored by local researchers, which is increasing in all three regions located in Asia, but stagnant in PAN. No research items indexed in WOS were authored by researchers from CAN (see Fig. 4).

### 3.2.4 International cooperation

Section 3.2.2 shows that items written by international research teams represent 29.1% of all GLOF research items with a gradually increasing trend. However, there are significant differences between individual countries (see Tab. 5). The highest number of international research items was authored (first author) by researchers from the UK (56 items), followed by the researchers from the USA (38). The researchers from the USA contributed to 91 international research items (36.1% of all), followed by researchers from UK who contributed to 84 international research items (33.3% of all). The share of international items on the overall number of items published among the top 10 GLOF research countries varies from 34.8% (Canada) to 71.4% (Norway).

The vast majority of the 45 GLOF research countries (n = 42) have contributed to at least one international GLOF research item (Lebanon, Saudi Arabia and Uzbekistan have not). The intensity of international cooperation between GLOF research countries is illustrated in Fig. 5. An overall number of 468 individual bilateral research ties exist (each tie represents one joint publication between two countries; see Tab. 1). Very strong cooperation (20+ joint research items) exists between researchers from the USA and the UK (24 joint research items), and the UK and Iceland (21), and strong cooperation (10+ joint research items) exists between the USA and Canada (19), the USA and Iceland (13), Switzerland and Germany (12), and the USA and Japan (10). Certain patterns and trends are observed regarding the cooperation between groups of countries and GLOF hotspots located in developing regions (see 4.2).

### 3.3 Analysis of the content

Selected aspects of the content and focus of GLOF research items published in the WOS Core Collection database are addressed in this section. A total of 823 GLOF research items (92.3%) are indexed as research articles (731 articles, 149 proceeding papers) and 50 items as reviews (5.6%). Additional 20 items contain "review*" within the title or abstract. A few items are also classified as letters, editorial materials or notes. Considering the different lake types that may be the source of GLOFs (moraine-dammed, bedrock-dammed, ice-dammed), it is shown that studies focusing on GLOFs originating in ice-dammed lakes (n = 451, 58.8%) dominate over studies focusing on GLOFs originating in moraine-dammed lakes (n = 195; 21.9%), roughly corresponding to the share of these lake types over the total number of GLOFs (see Carrivick and Tweed, 2016; Emmer et al., 2016).

In terms of distinguishing between studies focusing on palaeo-GLOFs (Pleistocene and Holocene) and recent ones (i.e. post-Little Ice Age), it is found that 258 GLOF research items (28.9%) explicitly focus on palaeo-events such as Lake Missoula and Lake Agassiz (see also 3.2.1). It is further stated in titles, abstracts and keywords, that 39.8% of GLOF research items (n = 355) use some kind of modelling approach / simulation, while 14.1% items (n = 126) explicitly declare the use of remotely-sensed data and 30.4% of GLOF research items (n = 271) deal with diverse aspects of GLOF hazard / risk assessment. Individual reviews of these selected aspects of GLOF research have been elaborated, e.g. by Huggel et al. (2002) or Quincey et al. (2005) focusing on the application of remotely-sensed data, or by Kougkoulos et al. (2018) focusing on GLOF susceptibility assessment, etc. (see also 4.3).

## 4 Observed trends in GLOF research: A discussion

### 4.1 Is there a changing publishing paradigm?

Research of glacial lake outburst floods (GLOFs) represents a dynamic research field, reflecting the needs and challenges brought by the rapidly changing environment (e.g., Huss et al., 2017). A certain change in the publishing paradigm is observed among the published GLOF research items and documented by the analysed characteristics (see Section 3.1). Firstly, the amount of research of GLOFs had an exponentially increasing trend in the WOS database between 1979 and 2016. From the field of natural hazard science, a similar trend is observed in research on tsunamis (Chiu and Ho, 2007), a more remarkable trend is observed in research on landslides (Xu et al., 2015) and a less remarkable trend in research on earthquakes (Liu et al., 2012). This trend is similar despite the fact that the GLOF research field is an order of magnitude smaller than the other above-mentioned fields and is in concordance with the generally observed trends across the scientific disciplines (e.g., Sandstrom and Van den Besselaar, 2016). Secondly, the increasing amount of published research items within the certain research field is directly tied with the increasing amount of citations obtained by individual items within this field (e.g., geomorphology; Dorn, 2002), which is also valid in the case of GLOF research (see 3.1.2). Thirdly, the share of GLOF research items written by individuals is observed to have dramatically declined over the few past decades (see 3.2.2). This observation is not surprising considering the general trends in research and science, and may indicate the increasing average extent and, hence, multidisciplinarity of the research teams involved (see also Skilton, 2009). Last but not least - the changing paradigm in GLOF research is also seen in the thematical content of GLOF research items. While hazards and risks, geomorphology and hydrology have traditionally dominated in GLOF research, recently other aspects such as climate justice have come to the forefront (e.g., Huggel et al., 2016; see 4.3).

### 4.2 Trends in geographies

A significant disproportion is observed between the number of documented GLOFs from specific regions and the number of GLOF research items geographically focused on a given region (see 3.2.1). While 270 GLOFs are documented from ICL (180 research items), only 47 are documented from HKH (142 research items; see Tab. 2). This disproportion can be explained as

a result of: (i) differences in the causes and mechanisms of GLOFs (repeated GLOFs from ice-dammed lakes in ICL, one-off GLOFs from moraine-dammed lakes in HKH); (ii) incomplete GLOF inventories in less researched and/or less settled regions such as remote areas of HKH (see Veh et al., 2018); (iii) the expected future increase in GLOF occurrence in HKH (Harrison et al., in review). On the other hand, the highest number of GLOFs are documented from ALP (n = 301), where only 46 studies were performed, which may be related to the late stage of glacier retreat in a post-LIA context (see also Emmer et al., 2015).

It has also been shown that an apparent geographical disproportionateness exists between the hotspots of GLOF occurrence (see 3.2.1) and top countries performing GLOF research (see 3.2.2), leading to the geographical "export" of research (see 3.2.3) - a phenomenon not yet fully captured within the context of the GLOF research field. Considering the items published in the WOS Core Collection database (see 2.1), the research of GLOFs is traditionally dominated by researchers from Europe and North America (see 3.2.2), while researchers from other countries - especially those overlapping with GLOF hotspots located in developing regions – have mostly focused on producing local publications and reports (see 2.1) and have come into play more recently, frequently as members of international research teams, but also as the first authors (Zaginaev et al., 2016; Colonia et al., 2017; Gherardini and Nucciotti, 2017; Prakash and Nagaraja, 2017). This trend can be explained by the increasing interest of local researchers in publishing research items indexed in WOS, instead of traditionally produced local publications and reports (see 2.1).

Strong collaboration is observed between the local institutions in developing regions and foreign researchers e.g. in Peruvian Andes, where none of the 11 research items (published until 2016) (co)authored by Peruvian researches have been led by them, but foreigner researchers. A similar but not so strong trend is also observed among the research items (co)authored by researchers from Central Asian countries (Kyrgyzstan, Kazakhstan). On the other hand, researchers from Nepal tend to publish their GLOF research items on their own - out of 20 GLOF research items (co)authored, 11 items were authored solely by the authors affiliated with Nepalese institutions (e.g., ICIMOD, Tribhuvan University). A similar trend is also observed for researchers in the PAN region (Argentina, Chile). The number of research items (co)authored by local researchers, however, significantly differs considering the total number of items focused on the given region (see 3.2.3, Fig. 4). A general trend of larger involvement of local authors and the internationalization of GLOF research teams has also been observed over the past few decades (see 3.2.4), which is in line with the general trends observed in global environmental change research (e.g., Jappe, 2007).

## 4.3 Challenges ahead and research directions

The research of GLOFs struggles with numerous challenges brought by the complexity of generic processes and the general characteristics of these events (low frequency, high magnitude, complicated predictability). One of the greatest challenges in the basic understanding of the spatio-temporal occurrence of GLOFs is to compile comprehensive databases of these events (see Carrivick and Tweed, 2016; Emmer et al., 2016). While past GLOFs are well-described in some of the regions (e.g., Alps; see also 3.2.1), others suffer from data scarcity (e.g., Veh et al., 2018). This different level of detail makes any assessment of

GLOF frequencies among the regions (on global level) a rather challenging task (Harrison et al., in review). Compiling regional databases of GLOFs aiming at a globally comprehensive database is, thus, considered one of the greatest challenges in GLOF research.

Since GLOFs have claimed thousands of lives and have caused considerable material damage in the past (Carrivick and Tweed, 2016), and further GLOFs are expected in the future reflecting the trend of retreating glaciers (Huss et al., 2017; Harrison et al., in review), hazard and risk assessment represents another highly challenging task and a recurrent topic among GLOF research items (see also 3.3). While hazard (susceptibility) assessment is traditionally dominant in GLOF risk studies (see the overview of methods by Kougkoulos et al., 2018), vulnerability assessment and adaptation in the broader context of retreating glaciers and water resources still remain brand new topics, which need to be further addressed in most regions and especially those located in developing countries (e.g., Vuille et al., 2018). Moreover, the communication of scientific results with local authorities and integration into the decision-making process is an extremely challenging task, which is far beyond the scope of geoscientific research (e.g., Gagne et al., 2014), promoting the need for broader interdisciplinary collaboration.

Various modelling tools are being developed and implemented hand in hand with the efforts to reliably assess the impacts of potential future GLOFs (e.g., Worni et al., 2014; Schaub et al., 2016; Chisolm and McKinney, 2017). Certain challenges are still seen ahead in this regard (e.g., Mergili et al., 2018) despite rapid progress in this direction and improvements to technical capabilities as well as data availability and acquirability (e.g., Mallalieu et al., 2017; Wigmore and Mark, 2017).

**5 Conclusions**

This study shows how research of glacial lake outburst floods (GLOFs) published in the Web of Science Core Collection database has become topical over the past few decades (analysed period 1979-2016); how the publishing culture and paradigm have changed over time and what the trends and disproportions in geographies of research on GLOFs are. A significant exponential increase in the number of GLOF research items published in the WOS database is revealed, with > 50% of the research items being published since 2008. While 1,885 researchers from more than 750 institutions in 45 countries have contributed to 892 of the analysed GLOF research items, a relatively small number of 90 leading researchers (4.8% of all) have published 5+ items each and have together contributed to almost half (48.9%) of all of the GLOF research items. Furthermore, the following trends were revealed among the published items over time: (i) internationalisation (increasing share of research items written by international teams); (ii) geographical disproportionateness (disparity between where GLOFs occurred and the top GLOF research countries). Eleven hotspots of GLOF occurrence and research have been identified, of which five are located in developing regions (South American Andes and high Asia). The export of research especially to these five hotspots has been documented hand in hand with the increasing involvement of local researchers in recent years. It was shown that researchers from the UK, the USA, Canada, Germany and Switzerland are the most active in exporting GLOF research worldwide and also in international cooperation. Despite the increasing amount of GLOF research items published

and the undisputable progress in this research field, numerous challenges in enhanced understanding GLOF and advanced risk management still remain ahead.

## Acknowledgement

I would like to thank Jonathan L. Carrivick for his insights into the early draft of this study, Petr Bašta for discussion on
processing Fig. 5, and Craig Hampson for language revision. I would also like to thank two anonymous referees for their comments and Sven Fuchs (NHESS Editor) for handling the manuscript. This work was supported by the Ministry of Education, Youth and Sports of CR within the National Sustainability Program I (NPU I), grant number LO1415.

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

Tab. 1: Analysed characteristics describing each GLOF item.

| Characteristics | Description | Value |
|---|---|---|
| **WOS database:** | | |
| Author(s) | A list of authors contributing to GLOF research items | 1,885 authors |
| Author's affiliation (institution) | A list of institutions contributing to GLOF research items | 764 / 787 institutions* |
| Author's affiliation (country) | A list of countries where the institutions are located | 45 countries |
| Year published | Year of publication in the journal | 1979**-2016 |
| Cited references in WOS collection | Number of references cited in one item | 0-280*** |
| Cited by | Cumulative number of citations obtained by one item | 0-581**** |
| Source title | Name of the journal / proceeding that published the GLOF research item | 256 different journals / proceedings |
| WOS categories | WOS category in which the journal / proceeding is indexed | 48 WOS categories |
| Research area | WOS research areas | 34 research areas |
| Document type | WOS document types | 9 document types |
| **Manually assigned characteristics (see 2.2):** | | |
| Geographical focus | Manually assigned geographical focus of the individual item, where relevant | 11 hotspots of GLOF occurrence; other regions |
| No. of authors | Items written by individuals are distinguished from items written by two or more authors (research teams) | Items written by an individual researcher; item written by a team |
| Mono-/inter-nationality | Items written by author(s) from one country (mononational) are distinguished from items written by authors from more than one country (international) | Item written by an individual researcher / mononational team; item written by an international team |
| International research ties | Manually assigned bilateral research ties (each tie represents one joint publication between two countries) | A matrix of 468 bilateral research ties |

* institutions / institutions-enhanced

** the first GLOF research item published in the WOS database (Jackson, 1979)

*** the highest number of references cited by one item (Girard et al., 2015)

5   **** the highest number of citations obtained by one item by the end of 2016 (Hemming, 2004)

Tab. 2: Hotspots of GLOF occurrence, number of documented GLOFs and GLOF research items focusing on a given hotspot.

| Hotspot of GLOF occurrence | Hotspot acronym | No. of documented GLOFs from a given region* | No. of GLOF research items geographically focusing on a given region | Examples of recent studies |
|---|---|---|---|---|
| **Asia:** | | | | |
| Central Asia (Pamir, Tien Shan) | CAS | 69 | 37 | Zaginaev et al., 2016; Petrov et al., 2017 |
| Hindu Kush - Himalaya (including Tibet) | HKH | 47 | 142 | Aggarwal et al., 2017; Nie et al., 2017 |
| Karakoram | KRK | 98 | 20 | Round et al., 2017 |
| **Europe:** | | | | |
| European Alps | ALP | 301 | 46 | Emmer et al., 2015 |
| Iceland | ICL | 270 | 180 | Guan et al., 2015 |
| Scandinavia | SCA | 121 | 21 | Xu et al., 2015 |
| **North America:** | | | | |
| Alaska | ALA | 80 | 34 | Wilcox et al., 2014 |
| Greenland | GRL | 22 | 34 | Carrivick et al., 2017; Grinsted et al., 2017 |
| North American Cordillera | NAC | 246 | 144 | Shaw et al., 2017 |
| **South America:** | | | | |
| Central Andes | CAN | 35 | 27 | Emmer, 2017 |
| Patagonian Andes | PAN | 92 | 30 | Iribarren Anacona et al., 2015 Wilson et al., 2018 |

* based on Carrivick and Tweed (2016); Falátková (2017); Emmer (2017); Wilson et al., 2018; note that the numbers of GLOF documented from less settled regions are likely to be underestimated (see also the text)

Tab. 3. Research exports from the top 10 countries to 11 hotspots of GLOF occurrence. Only the reprint address of the first author is considered in this table (each item is counted only once). The number of items from overlapping countries (OC; e.g., Switzerland and ALP; Norway and SCA) is shown in brackets.

| Country | CAS | HKH | KRK | ALP | ICL | SCA | ALA | GRL | NAC | CAN | PAN | Excl. OC | | Incl. OC | |
|---|---|---|---|---|---|---|---|---|---|---|---|---|---|---|---|
| | \multicolumn Hotspots of GLOFs occurrence (see also Tab. 2) | | | | | | | | | | | Total | | | |
| TG | 37 | 142 | 20 | 46 | 180 | 21 | 34 | 34 | 144 | 27 | 30 | 715 | | | |
| USA | 1 | 19 | 1 | 1 | 15 | 1 | (26) | 8 | (61) | 5 | 1 | 52 | 2. | 139 | 1. |
| UK | 4 | 13 | 1 | 5 | 76 | 4 | 2 | 12 | 9 | 4 | 6 | 136 | 1. | 136 | 2. |
| Canada | 1 | 2 | 2 | 2 | 4 | 0 | 3 | 2 | (61) | 1 | 0 | 17 | 6. | 78 | 3. |
| Iceland | 0 | 0 | 0 | 0 | (49) | 0 | 0 | 0 | 0 | 0 | 0 | 0 | 10. | 49 | 4. |
| Germany | 5 | 15 | 4 | (1) | 8 | 0 | 0 | 1 | 2 | 1 | 3 | 39 | 3. | 40 | 6. |
| China | (6) | (16) | (3) | 0 | 0 | 1 | 0 | 0 | 0 | 0 | 0 | 1 | 8./9. | 26 | 8. |
| Switzerland | 3 | 7 | 2 | (19) | 1 | 0 | 0 | 0 | 0 | 5 | 2 | 20 | 5. | 39 | 5. |
| Japan | 3 | 21 | 1 | 2 | 0 | 0 | 1 | 0 | 1 | 0 | 0 | 29 | 4. | 29 | 7. |
| France | 0 | 1 | 0 | 9 | 6 | 0 | 0 | 0 | 0 | 0 | 0 | 16 | 7. | 16 | 9. |
| Norway | 0 | 1 | 0 | 0 | 0 | (7) | 0 | 0 | 0 | 0 | 0 | 1 | 8./9. | 8 | 10. |
| TI10 | 17 | 79 | 11 | 19 | 110 | 6 | 6 | 23 | 12 | 16 | 12 | 311 | | 560 | |
| TI10/1G | 45.9 | 55.6 | 55.0 | 41.3 | 61.1 | 28.6 | 23.1 | 67.6 | 8.3 | 59.3 | 40.0 | 43.5 | | 78.3 | |

TG - Total number of items geographically focused on a given region; TI10 - Total number of items authored by researchers from the top 10 GLOF research countries (excluding overlapping countries)

Tab. 4. Geographical focus of research items of the top 10 GLOF research countries (11 hotspots of GLOF occurrence). All items (co)authored by researchers from the given country are considered in this table (i.e. research items written by international research teams are counted for each country separately). The number of items from overlapping countries (OC; e.g., Switzerland and ALP; Norway and SCA) is shown in brackets.

| Country | Hotspots of GLOFs occurrence (see also Tab. 2) | | | | | | | | | | | Total | | | |
| --- | --- | --- | --- | --- | --- | --- | --- | --- | --- | --- | --- | --- | --- | --- | --- |
| | CAS | HKH | KRK | ALP | ICL | SCA | ALA | GRL | NAC | CAN | PAN | Excl. OC | | Incl. OC | |
| TG | 37 | 142 | 20 | 46 | 180 | 21 | 34 | 34 | 144 | 27 | 30 | 715 | | | |
| USA | 3 | 27 | 4 | 1 | 30 | 1 | (28) | 11 | (76) | 6 | 1 | 84 | 2. | 188 | 1. |
| UK | 6 | 19 | 1 | 7 | 82 | 4 | 3 | 12 | 9 | 4 | 7 | 154 | 1. | 154 | 2. |
| Canada | 2 | 7 | 3 | 3 | 5 | 0 | 4 | 5 | (70) | 1 | 2 | 32 | 6. | 102 | 3. |
| Iceland | 0 | 0 | 0 | 0 | (78) | 0 | 0 | 0 | 0 | 0 | 0 | 0 | 10. | 78 | 5. |
| Germany | 8 | 17 | 4 | (2) | 10 | 0 | 0 | 3 | 3 | 2 | 4 | 51 | 4. | 53 | 6. |
| China | (11) | (22) | (6) | 0 | 1 | 1 | 0 | 1 | 0 | 0 | 0 | 3 | 9. | 42 | 7. |
| Switzerland | 7 | 15 | 5 | (27) | 3 | 2 | 2 | 2 | 4 | 9 | 4 | 53 | 3. | 80 | 4. |
| Japan | 3 | 26 | 1 | 3 | 0 | 0 | 1 | 0 | 2 | 0 | 0 | 36 | 5. | 36 | 8. |
| France | 0 | 2 | 0 | (11) | 9 | 0 | 0 | 0 | 0 | 0 | 1 | 12 | 8. | 23 | 10. |
| Norway | 3 | 5 | 0 | 1 | 3 | (8) | 0 | 3 | 3 | 0 | 0 | 18 | 7. | 26 | 9. |

5   TG - Total number of items geographically focusing on a given region

Tab. 5. International research items authored and co-authored by researchers from the top 10 GLOF research countries.

| | USA | UK | Canada | Iceland | Germany | China | Switzerland | Japan | France | Norway |
|---|---|---|---|---|---|---|---|---|---|---|
| Total number of research items (co)authored | 240 | 213 | 115 | 85 | 68 | 65 | 60 | 38 | 30 | 28 |
| No. of international items authored | 38 | 56 | 14 | 17 | 22 | 10 | 19 | 13 | 4 | 1 |
| No. of international items co-authored | 53 | 28 | 26 | 34 | 17 | 16 | 23 | 10 | 9 | 19 |
| No. of international items (authored + co-authored) | 91 | 84 | 40 | 51 | 39 | 26 | 42 | 23 | 13 | 20 |
| % share on the total number of research items written by an international team (n = 252) | 36.1 | 33.3 | 15.9 | 20.2 | 15.5 | 10.3 | 16.7 | 9.1 | 5.2 | 7.9 |
| % share on the total number of research items of the given country | 37.9 | 39.3 | 34.8 | 60.0 | 57.4 | 40.0 | 70.0 | 60.5 | 43.3 | 71.4 |

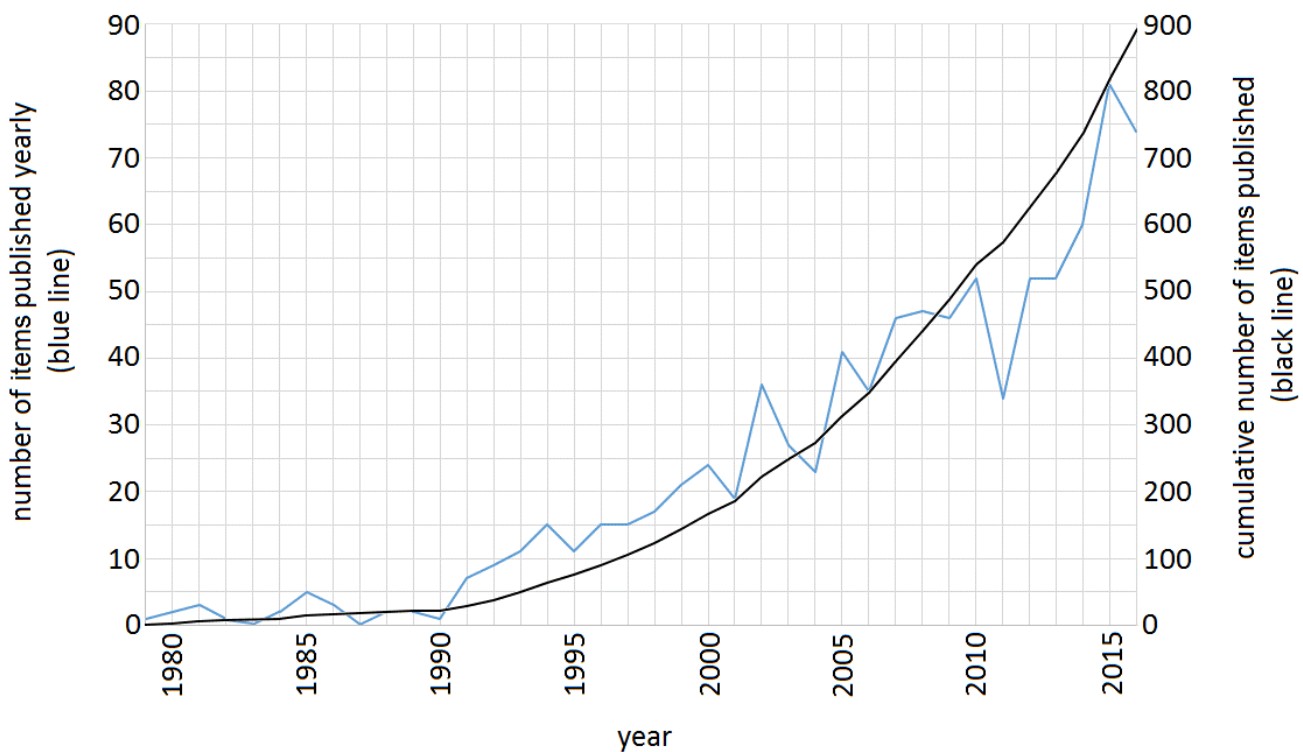

**Figure 1: Yearly and cumulative number of GLOF research items published in the WOS Core Collection database (the first item was published in 1979).**

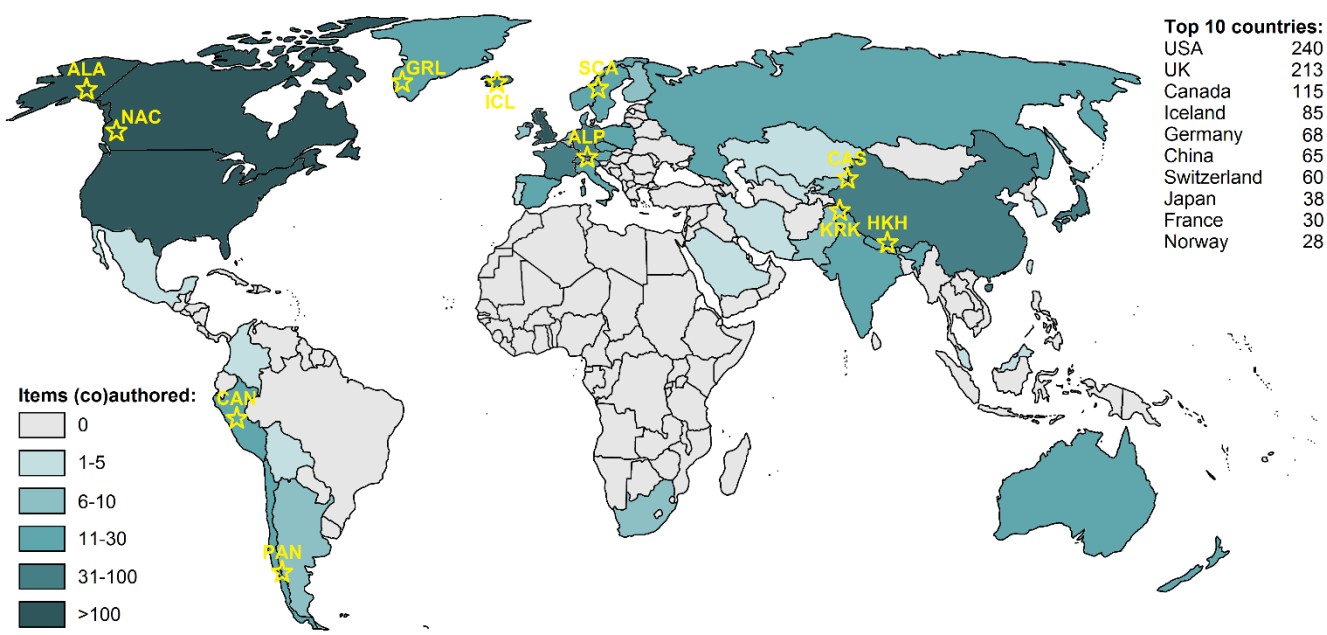

Top 10 countries:

| | |
|---|---|
| USA | 240 |
| UK | 213 |
| Canada | 115 |
| Iceland | 85 |
| Germany | 68 |
| China | 65 |
| Switzerland | 60 |
| Japan | 38 |
| France | 30 |
| Norway | 28 |

Items (co)authored:
- 0
- 1-5
- 6-10
- 11-30
- 31-100
- >100

**Figure 2: Hotspots of GLOF occurrence (yellow stars; see Tab. 2 for hotspot acronyms) and the geographical distribution of published GLOF research items by country.**

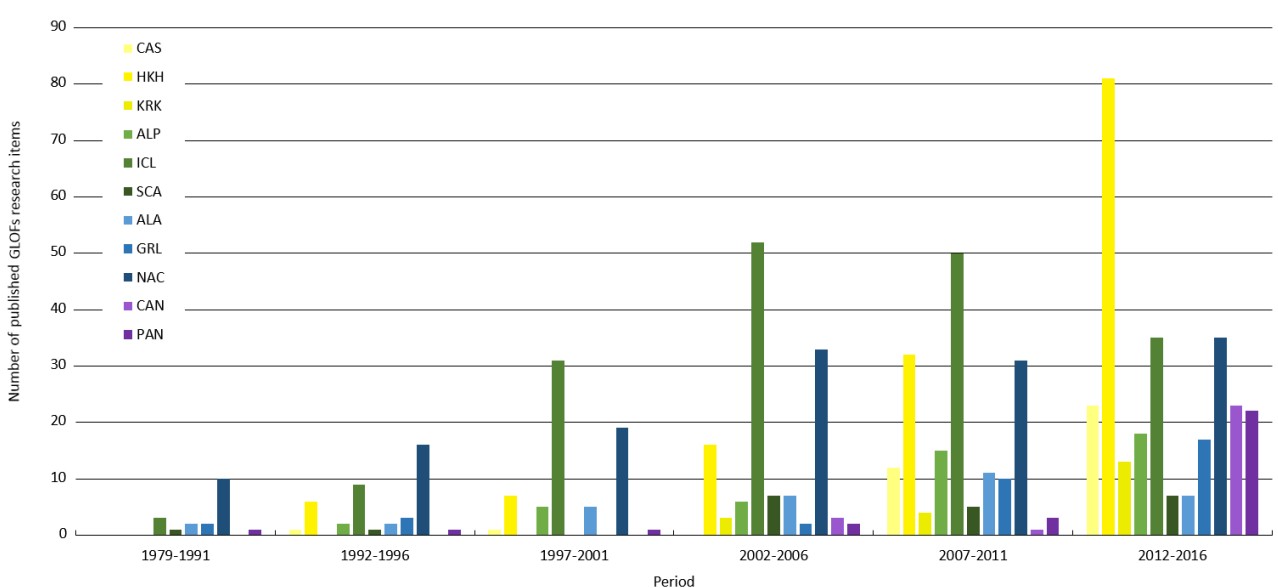

**Figure 3: The amount of GLOF research items focusing on individual GLOF research hotspots (see Fig. 2) over time.**

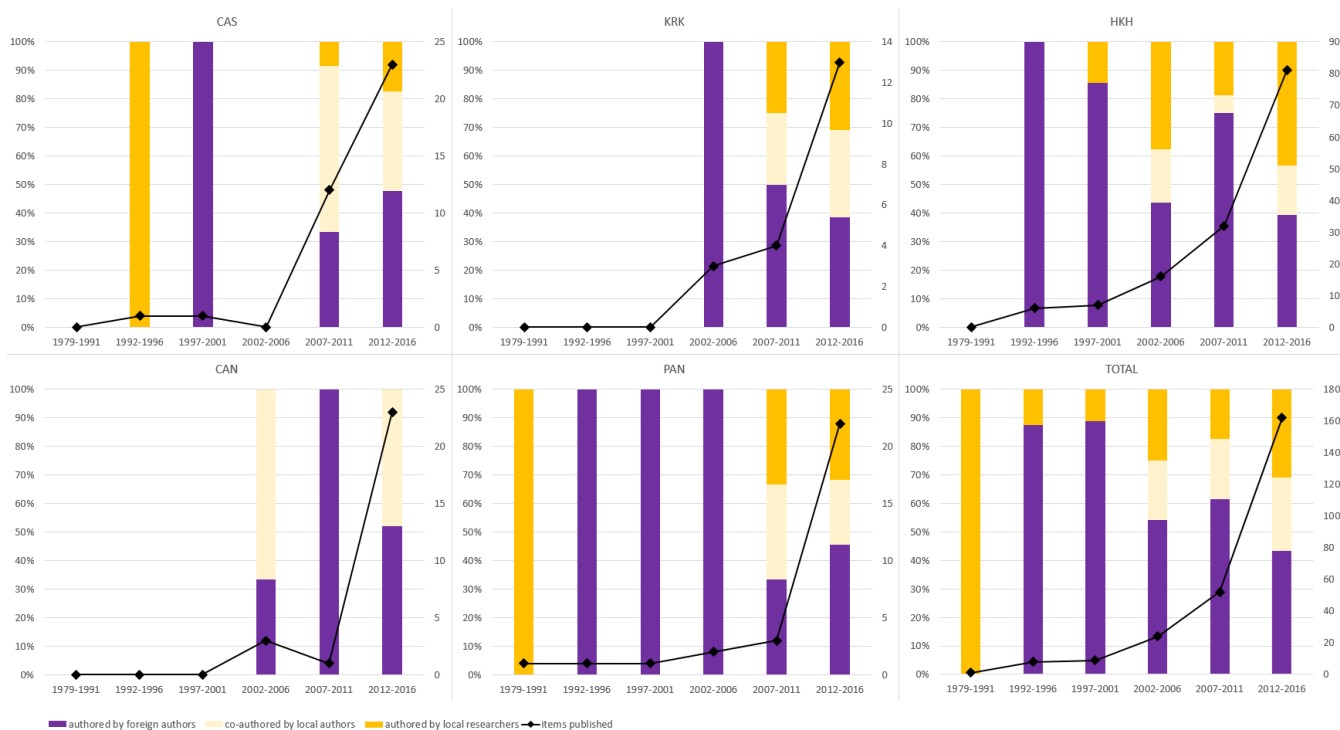

**Figure 4: GLOF research items geographically focusing on five research hotspots located in developing regions (see also Figs. 2, 3). The involvement of local researchers is shown. Note the different scales of y axis for the number of GLOF research items published (black line).**

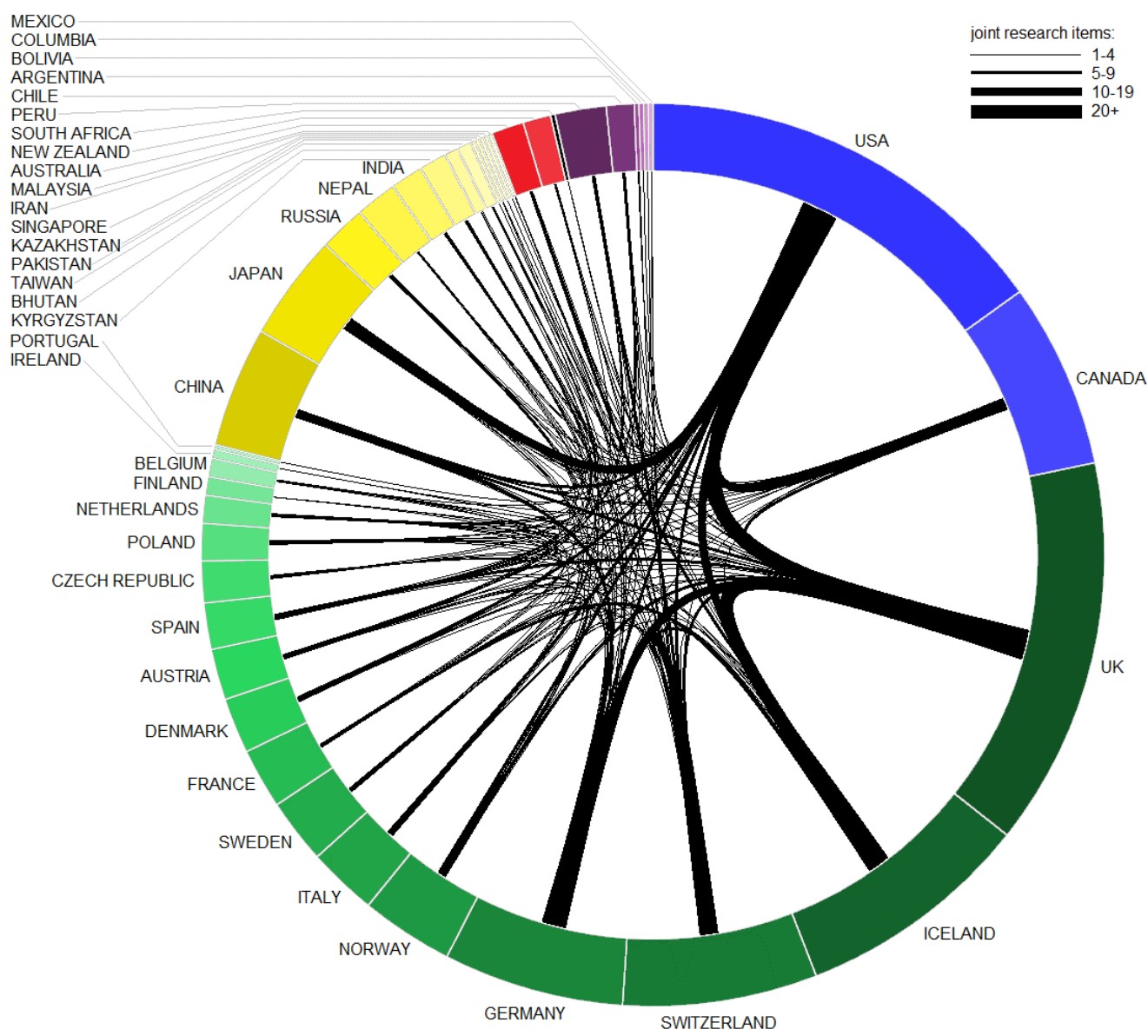

**Figure 5: International cooperation between GLOF research countries. The number of international items (co)authored is proportionally indicated by the size of a sector (see also Tab. 5).**

