# Peer review of "GLOFs in the WOS: bibliometrics, geographies and global trends of research of glacial lake outburst floods (Web of Science, 1979-2016)"

_Natural Hazards and Earth System Sciences, 2017_

## Referee Comment (RC1) · Anonymous Referee #1 · 5 Jan 2018

nhess-2017-413 GLOFs in the WOS: bibliometrics, geographies and global trends of research of glacial lake outburst floods (Web of Science, 1979-2016) Adam Emmer

General Comments: The paper analyzes 892 GLOF research items published in the Web of Science database during the period 1979-2016. A change in the publishing paradigm over time is proposed and global geographies of research on GLOFs were studied.

The paper is a good analysis of the publication database related to GLOFs and might make a good technical note or editorial in NHESS or in a journal devoted to academic publication trends in the geosciences. It is beyond my expertise to say if this paper

deserves publication as a research paper in NHESS. I do not find information in the paper that enhances my research on GLOFs.

In the instructions to reviewers, we are asked to address some specific questions, specific ones that I find relevant to this paper are:

1. Does the paper address relevant scientific and/or technical questions within the scope of NHESS? 2. Does the paper present new data and/or novel concepts, ideas, tools, methods or results?

I cannot respond positively to these questions regarding this paper. In particularly, scientific or technical questions realted to GLOFs are not presented or discussed. Certainly, Web of Science data related to publications in various journals is presented, but this is not related to the scientific/technical questions of the field.

Specific Comments:

P2-L1: Actually glacial lake outburst flood is in the title of Jackson's 1979 paper: Jackson, L. E.: Catastrophic glacial lake outburst flood (jokulhlaup) mechanism for debris flow generation at the spiral tunnels, Kicking horse River basin

P2-L2: "Clarke and Mathews (1981)" reference missing

---

## Referee Comment (RC2) · Anonymous Referee #2 · 16 Jan 2018

Dear Editor,

Thank you for inviting me to review this manuscript. Author proposed the manuscript "GLOFs in the WOS: bibliometrics, geographies, and global trends of research of glacial lake outburst floods (Web of Science, 1979-2016)". The analysis of this paper is interesting, but some parts should be improved and think more deeply. I do not know that this paper is suitable for this journal, because this paper does not include scientific research and detail discussion. I still do not understand a benefit information which author would like to show in this paper. Readers of NHESS might be more interested in regional differences of GLOF including the characteristics of GLOF, research

themes, and research methods. In addition, it is difficult to show the trend using only Web of Science, because author does not treat local publications.

General comments: 1) It is not clear what this paper will be useful for future GLOF research. What is benefit information for GLOF studies? Please indicate what your suggestions in the future GLOF research, because I cannot confirm your opinion in the manuscript based on data.

2) The tendency of the study area where GLOF occurred and the foreigners were active varies each decade and region. Based on analysis each decade (e.g. between 1979-1999 and 2000-2016), author might understand the transition of the region. Although author analyzed about time series is whether it is a single author or co-authorship, readers do not have great interest about this topic.

4) In this paper, author used the data of Web of Science, but there are many local papers in local journal. For example, it is often written in the USSR. Tables 3 and 4 do not include Russia or USSR. The paradigm shift is clearly a mistake, because this paper is intended for 1979-2016. In USSR territory, there are overwhelmingly local publications before 2000. Please indicate time series each region and mention regions which foreigners wrote mainly (where no paradigm shift has occurred).

5) In HKT, as a representative of local researchers, I think that contribution such as local scientist of ICIMOD is very large, but there is no mention on that point about paradigm shift. One sentence is not enough for explanation of paradigm shift in HKT. This topic should be written more deeply in discussion.

Specific comments and technical notes:

Page 1 L10-L11: What is GLOF research items? Definition of GLOF paper is not clear. For example, glacier lake research (simply changes of lake area) is included? Although theme of papers is glacier changes, there are the cases which these include term of GLOF. Did you confirm each article?

L12: Paradigm over time. . .. Please show the paradigm shift (time series) based on each regional study and activity including local publication, because only data base of Web of Science does not show the correct regional trend.

L18: dam failure and dam overtopping, or glacial lake sub-type?

Page 2 L30: Although articles include term of GLOF, some articles are not theme of GLOF. Did you confirm each article?

Page 6 L7-14: Trends are also different each developing country. You should write concretely the regionality. For example, there are no differences about foreigner activities for Central Asia and Karakoram, but Himalaya are the most common in US and Japan. The contribution of these countries is different each decade since 1979. Based on summary of developing country, it is impossible to see individual details and transition.

Page 9 L2-10: I cannot agree with your opinion. Central Asia and Caucasus have been active mainly by many local researchers since the USSR. They published many articles about GLOF in Russian journal. There is little description of foreigners. In addition to data base of Web of Science, author should use local publications. An example in Himalaya is not appropriate. In Himalaya, we should state the activities of ICIMOD.

L24: Please analyze each region of the developing country.

L25: I think recent local researchers are active in HKT.

---

## Author Response (AR1)

Dear Dr. Sven Fuchs, dear referees,

Thank you for giving me the chance to revise my manuscript „GLOFs in the WOS: bibliometrics, geographies and global trends of research of glacial lake outburst floods (Web of Science, 1979-2016)" submitted to NHESSD. I appreciate the comments and suggestions of referees and tried to revise the manuscript accordingly.

Most importantly, following changes have been made:

(i) Extended discussion on significance of local publications and reports not indexed at WOS, especially in section 2.1 but also in other relevant sections;

(ii) New section *3.3 Analysis* of the *content* providing basic overview on focus of GLOF research items and *4.3 Challenges ahead and research directions* have been introduced;

(iii) Sections 3.2.1 and 3.2.3 have been extended by more detailed analysis of GLOF research activities in space and time with special focus on GLOF research hotspots located in developing regions;

(iv) Section 4.2 has been reworked in line with more detailed findings of extended sections 3.2.1 and 3.2.3;

(v) New figures 3 and 4 have been produced;

(vi) The list of references has been updated by recently published literature.

Point-by-point answers to the reviews as well as revised manuscript with all the changes visible in changes-tracking mode are attached below.

Thank you, kind regards

Adam Emmer

**Point-by-point reply to reviews:**

**General response:**
I'd like to thank both referees for their time, comments and suggestions. In my general response, I'd like to tackle three major issues:

**(1) the choice of the journal**
Both reviewers are not sure about the suitability of NHESS for publishing such work. Considering the focus of the manuscript, I fully undestand this concern. When I have prepared the manuscript, I've had a hard time deciding where to submit. There were two options in terms of research areas: (i) information science (scientometrics / bibliometrics); or (ii) natural hazard science. As it is shown in Introduction of the manuscript - numerous scientometric / bibliometric studies focusing on various fields of natural hazard science (landslides, tsunamis, ...) already exist, some of them published in scientometric journals and some of them in natural hazard science journals. In the case of my manuscript, I'm convinced that natural hazard science research community might be of higher interest and publishing in journal in natural hazard science area might provide higher visibility, therefore I've decided for NHESS.

**(2) scientific content and novelty**
I'd like to stress, that this contribution does not primarily aim at providing new information about GLOF and its generic physical processes (as pointed out by Referee #2), but about research on GLOFs, its bibliometrics, trends and geographies (where GLOFs are studied; who studies GLOFs; the export of research on GLOFs; and international collaboration). Compared to other types of natural hazards (e.g., landslides, tsunamis, earthquakes) this topicis not yet addressed and, thus, novel. This study shows how publishing culture and paradigm have changed over time, based on analysis of GLOF research items published at WOS (see also point (3)). From this perspective, I'm convinced that new findings and results which might be of interest for GLOF research community as well as practitioners, are presented.

**(3) the use of WOS database**
The use of WOS database is justified in Section 2.1. Certainly, I'm aware of local publications (see also Introductio and section 2.1), however analysing local publications and aiming on global perspective at the same time is far beyond objectives, scope and resources available for this study. Moreover, based on my experience from Andes, local publications are often not available online (only available in local libraries). Results of presented work are based on research items indexed in the WOS database and, thus, only valid for WOS data, which is clearly stated in the manuscript. I'm, however, still convinced, that the study provides valuable insights into the GLOF research community and geographies of GLOF research (see above).

In line with the suggestions and recommendations of two referees, following major changes will be done in the revised version of the manuscript:

(i) deeper discussion on limits of WOS database and potentials of local publications;
(ii) deeper investigation of trends on regional level in time; and
(iii) discussion of suggestions for future GLOF research.

Point-by-point replies are shown below (in blue) with recent updates compared to open discussion (in green).

Thank you, kind regards

Adam Emmer

**Anonymous Referee #1**

General Comments: The paper analyzes 892 GLOF research items published in the Web of Science database during the period 1979-2016. A change in the publishing paradigm over time is proposed and global geographies of research on GLOFs were studied. The paper is a good analysis of the publication database related to GLOFs and might make a good technical note or editorial in NHESS or in a journal devoted to academic publication trends in the geosciences. It is beyond my expertise to say if this paper deserves publication as a research paper in NHESS. I do not find information in the paper that enhances my research on GLOFs.

- Please see my general comments
- The use of WOS database and the potential of local publications and reports is elaborated in more detail in revised version of the manuscript in section 2.1

In the instructions to reviewers, we are asked to address some specific questions,
specific ones that I find relevant to this paper are:
1. Does the paper address relevant scientific and/or technical questions within the scope of NHESS?
2. Does the paper present new data and/or novel concepts, ideas, tools, methods or results?
I cannot respond positively to these questions regarding this paper. In particullary, scientific or technical questions realted to GLOFs are not presented or discussed. Certainly, Web of Science data related to publications in various journals is presented, but this is not related to the scientific/technical questions of the field.

- I understand this concern, please see my general comments
- The extent of the study has been broadened and new sections introduced in revised version of the manuscript

Specific Comments:
P2-L1: Actually glacial lake outburst flood is in the title of Jackson's 1979 paper: Jackson, L. E.: Catastrophic glacial lake outburst flood (jokulhlaup) mechanism for debris flow generation at the spiral tunnels, Kicking horse River basin

- According to the WOS database and the publisher website http://www.nrcresearchpress.com/doi/abs/10.1139/t79-087#.Wl8kYq7ibIU the author did not use 'lake' in title or abstract, just 'glacial outburst flood'
- Checked, no change

P2-L2: "Clarke and Mathews (1981)" reference missing

- will be added
- is added in revised version of the manuscript

**Anonymous Referee #2**

Dear Editor,

Thank you for inviting me to review this manuscript. Author proposed the manuscript "GLOFs in the WOS: bibliometrics, geographies, and global trends of research of glacial lake outburst floods (Web of Science, 1979-2016)". The analysis of this paper is interesting, but some parts should be improved and think more deeply. I do not know that this paper is suitable for this journal, because this paper does not include scientific research and detail discussion. I still do not understand a benefit information which author would like to show in this paper. Readers of NHESS might be more interested in regional differences of GLOF including the characteristics of GLOF, researchthemes, and research methods. In addition, it is difficult to show the trend using only Web of Science, because author does not treat local publications.

- See also my general comment (3); the potential of local publications will be discussed in more detail

- The use of WOS database and the potential of local publications and reports is elaborated in more detail in revised version of the manuscript in section 2.1 and other relevant parts of the manuscript
- Selected aspects of the content of published GLOF research items are analysed in newly introduced section 3.3
- Challenges ahead and research directions are elaborated in section 4.3

General comments:

1) It is not clear what this paper will be useful for future GLOF research. What is benefit information for GLOF studies? Please indicate what your suggestions in the future GLOF research, because I cannot confirm your opinion in the manuscript based on data.

- Separate section on suggestions for future GLOF research will be introduced in the revised version of the manuscript
- Newly introduced section 4.3 discus this issue

2) The tendency of the study area where GLOF occurred and the foreigners were active varies each decade and region. Based on analysis each decade (e.g. between 1979-1999 and 2000-2016), author might understand the transition of the region. Although author analyzed about time series is whether it is a single author or co-authorship, readers do not have great interest about this topic.

- More detailed analysis will be included in revised version of the manuscript
- More detailed spatio-temporal analysis especially focusing on GLOF hotspots located in developing regions are presented in sections 3.2.1 and 3.2.3 in revised version of the manuscript

4) In this paper, author used the data of Web of Science, but there are many local papers in local journal. For example, it is often written in the USSR. Tables 3 and 4 do not include Russia or USSR. The paradigm shift is clearly a mistake, because this paper is intended for 1979-2016. In USSR territory, there are overwhelmingly local publications before 2000. Please indicate time series each region and mention regions which foreigners wrote mainly (where no paradigm shift has occurred).

- again, only research items published at WOS database are analysed in presented study; see also my general response
- this part of the section 4.2 has been reworked using more detailed detailed spatio-temporal analysis especially focusing on GLOF hotspots located in developing regions (sections 3.2.1 and 3.2.3)

5) In HKT, as a representative of local researchers, I think that contribution such as local scientist of ICIMOD is very large, but there is no mention on that point about paradigm shift. One sentence is not enough for explanation of paradigm shift in HKT. This topic should be written more deeply in discussion.

- I agree with this point; this part will be elaborated in more detail in revised version of the manuscript
- More detailed spatio-temporal analysis especially focusing on GLOF hotspots located in developing regions are presented in sections 3.2.1 and 3.2.3 in revised version of the manuscript

Specific comments and technical notes:

Page 1 L10-L11: What is GLOF research items? Definition of GLOF paper is not clear. For example, glacier lake research (simply changes of lake area) is included? Although theme of papers is glacier changes, there are the cases which these include term of GLOF. Did you confirm each article?

- „research item" is WOS term referring to individual record in the WOS database; thses are mainly papers / articles, but also book chapters, conference papers and others (see also Section 3.1.1)
- Only those items found using carefully designed searching formulae are analysed; glacier lake research is not analysed if the abstract / title does not explicitly mention glacial lake outburst / GLOF
- GLOF research item is described in section 2.1 and its characteristics in Tab. 1

L12: Paradigm over time: : :. Please show the paradigm shift (time series) based on each regional study and activity including local publication, because only data base of Web of Science does not show the correct regional trend.

- I'm aware of local publications (see also section 2.1), however analysing local publications and aiming on global perspective at the same time is far beyond objectives, scope and resources of this study (see also my general comments)
- using the WOS database shows (correct) regional trend among WOS publications, which is stress several times in the manuscript as well as obvious from the title
- this issue will be discussed in more detail in the revised version of the manuscript
- The use of WOS database and the potential of local publications and reports is elaborated in more detail in revised version of the manuscript in section 2.1; the use of WOS database is stress in the several relevant places in the manuscript

L18: dam failure and dam overtopping, or glacial lake sub-type?

- „irrespective of the cause (trigger), mechanism (dam failure or dam overtopping) or glacial lake sub-type involved"
- edited („or" replaced by „and")

Page 2 L30: Although articles include term of GLOF, some articles are not theme of GLOF. Did you confirm each article?

- In general, yes. Firstly, the search formulae was carefully defined to select items dealing with GLOFs, using a set of articles for double check; I, honestly, did not go through all the full texts of 892 GLOF research items found, but the abstracts; and I can confirm that each item touches GLOFs, at least partially.
- Even if some of the 892 items considered in the analysis would not dealt with GLOFs, this would not changed the overall figure and trends observed

Page 6 L7-14: Trends are also different each developing country. You should write concretely the regionality. For example, there are no differences about foreigner activities for Central Asia and Karakoram, but Himalaya are the most common in US and Japan. The contribution of these countries is different each decade since 1979. Based on summary of developing country, it is impossible to see individual details and transition.

- This section will be elaborated in more detail in revised version of the manuscript
- More detailed spatio-temporal analysis especially focusing on GLOF hotspots located in developing regions are presented in sections 3.2.1 and 3.2.3 in revised version of the manuscript
- Please see newly introduced Fig. 4

Page 9 L2-10: I cannot agree with your opinion. Central Asia and Caucasus have been active mainly by many local researchers since the USSR. They published many articles about GLOF in Russian journal. There is little description of foreigners. In addition to data base of Web of Science, author should use local publications. An example in Himalaya is not appropriate. In Himalaya, we should state the activities of ICIMOD.

- I agree with the reviewer that the use of local publications might be beneficial, there are, however, numerous obstacles which make it hardly feasible on global level (see also my general comments)
- The use of WOS database and the potential of local publications and reports is elaborated in more detail in revised version of the manuscript in section 2.1 and other relevant parts of the manuscript
- I fully agree that ICIMOD did a lot of work in HKH region as well as USSR researchers in Central Asia and the importance of local publications will be discussed in more details in revised version of the manuscript
- The activities of ICIMOD are also already covered in the analysis (the researchers from ICIMOD have contributed to 9 research items indexed in the WOS database)
- This is discussed in more detail in revised version of the manuscript

L24: Please analyze each region of the developing country.
- This part will be extended as suggested by the referee
- More detailed spatio-temporal analysis especially focusing on GLOF hotspots located in developing regions are presented in sections 3.2.1 and 3.2.3 in revised version of the manuscript

L25: I think recent local researchers are active in HKT.
- Will be checked and eventually rewritten
- More detailed spatio-temporal analysis especially focusing on GLOF hotspots located in developing regions are presented in sections 3.2.1 and 3.2.3 in revised version of the manuscript
- Please see newly introduced Fig. 4

[revised manuscript text omitted]